# Understanding Students’ Vaccination Literacy and Perception in a Middle-Income Country: Case Study from Kazakhstan

**DOI:** 10.3390/vaccines12080917

**Published:** 2024-08-15

**Authors:** Karina Nukeshtayeva, Nurbek Yerdessov, Olzhas Zhamantayev, Aliya Takuadina, Gaukhar Kayupova, Zhaniya Dauletkaliyeva, Zhanerke Bolatova, Ganisher Davlyatov, Aizhan Karabukayeva

**Affiliations:** 1School of Public Health, Karaganda Medical University, Gogol Street 40, Karaganda 100008, Kazakhstan; nukeshtaeva@qmu.kz (K.N.); kayupovag@qmu.kz (G.K.); dauletkalievaz@qmu.kz (Z.D.); bolatovazhanerke93@gmail.com (Z.B.); 2Department of Informatics and Biostatistics, Karaganda Medical University, Gogol Street 40, Karaganda 100008, Kazakhstan; takuadina@qmu.kz; 3Hudson College of Public Health, The University of Oklahoma Health Sciences, Oklahoma City, OK 73104, USA; ganisher-davlyatov@ouhsc.edu (G.D.); aizhan-karabukayeva@ouhsc.edu (A.K.)

**Keywords:** vaccination, vaccines, immunization, perception, literacy, attitude, university, student, Kazakhstan, middle-income country

## Abstract

Vaccination is a critical public health measure for preventing infectious diseases, but its acceptance varies globally, influenced by factors like vaccine hesitancy. This study examines attitudes and vaccination literacy among Kazakh students, providing insights into global immunization strategies. A cross-sectional survey was conducted with 3142 students from various Kazakh universities. The HLS19-VAC instrument assessed vaccination literacy, while additional questions evaluated beliefs and attitudes toward vaccines. Data were analyzed to determine associations between vaccine-related beliefs and literacy. The mean vaccination literacy score was 84.74. Most students agreed on the importance (83.4%), safety (79.1%), effectiveness (80.9%), and religious compatibility (77.8%) of vaccines. Positive beliefs significantly correlated with higher literacy scores. Past vaccination behavior, age, gender, and location showed varied associations, with past vaccination status and higher age showing a positive correlation. Positive beliefs about vaccinations were strongly associated with higher vaccination literacy among Kazakh students. Educational interventions that reinforce positive beliefs may improve vaccination literacy and increase vaccination rates. This study underscores the importance of understanding vaccination attitudes to enhance public health strategies in middle-income countries.

## 1. Introduction

Vaccination, a highly successful public health measure against infectious diseases, has become the focus of intensified global discourse, particularly due to the COVID-19 pandemic and outbreaks of vaccine-preventable diseases [1,2]. Despite its benefits, public acceptance varies, influenced by vaccine hesitancy [3]. From 2015 to 2017, over 90% of WHO member states reported vaccine hesitancy, and in 2019, it was listed as a top ten global health challenge [4]. While high-income countries often dominate this discourse, middle- and low-income countries face unique challenges and opportunities. This study explores vaccination literacy and attitudes among students in Kazakhstan, contributing to the broader understanding of global immunization strategies.

People’s ability to find, understand, and use health information, a concept referred to as health literacy, is seen to be critical in judgements and decisions about health care, prevention, and health promotion [5]. A decision to get vaccinated or not is a health behavior that could be influenced by various factors, including general concerns about vaccine safety and efficacy, as well as convenience and timing [6]. Vaccination Health Literacy (VHL), a component of general health literacy, significantly impacts vaccination decisions. Biasio et al. and Gusar et al. found that vaccination literacy stems from health literacy, involving the motivation and ability to understand and use vaccine information [7,8]. Studies showed that higher VHL predicts higher vaccine completion rates, such as for HPV in the U.S. and COVID-19 among nursing students in Saudi Arabia [9,10]. Increased VHL is associated with better vaccine uptake and compliance, reducing vaccine-preventable diseases [11].

Attitudes toward vaccination vary globally, influenced by cultural, religious, socioeconomic, and political factors. While vaccination is recognized as effective, acceptance rates differ by region [12]. In many high-income countries, vaccination is generally well-accepted, supported by strong healthcare systems, robust public health campaigns, and high levels of trust in medical authorities. However, recent findings indicate that full immunization coverage is sometimes lower among wealthier populations compared to poorer ones in certain countries, particularly within the upper-middle income group, reflecting the emergence of vaccine hesitancy [13]. This phenomenon has been linked to outbreaks of vaccine-preventable diseases, such as measles, emphasizing the need to address vaccine hesitancy even in nations with well-established immunization programs [14]. For instance, the measles outbreak in Kazakhstan in 2023 can be primarily attributed to the accumulation of susceptible children who missed their scheduled immunization doses during the COVID-19 pandemic [15]. In contrast, low- and middle-income countries face additional challenges like access, affordability, and infrastructure. Mistrust in vaccine safety is particularly pronounced in low-income countries, where misinformation and lack of reliable health information can deter vaccination [16].

According to the Decree of the Government of the Republic of Kazakhstan, routine preventive vaccinations are administered in accordance with the National Immunization Calendar of the Republic of Kazakhstan. Mandatory vaccinations include those against tuberculosis, hepatitis, polio, whooping cough, diphtheria, tetanus, Haemophilus influenza type b, pneumococcal infection, measles, rubella, and mumps [17]. All mandatory vaccinations are provided free of charge and funded by the government. Since 2016, the vaccination coverage of the population, as per the National Immunization Calendar in Kazakhstan, has exceeded 80% [18].

The results of vaccination attitude surveys in Kazakhstan demonstrated an interesting and primarily positive dynamic. In November 2020, Demoscope’s survey indicated critically low levels of anti-COVID vaccination acceptance in Kazakhstan—around 45% of surveyed did not want to receive any vaccine, 33.33% wished to wait, and only 14% expressed the desire to be vaccinated as soon as possible [19].

A survey by the Public Opinion Research Institute supported Demoscope’s findings, showing worsening trends: 55.5% were hostile toward vaccination, and only 22.9% supported it [20]. Conducted at the end of 2020, the survey indicated a negative sentiment. However, a Demoscope survey in early May showed a significant positive shift, with 59% favoring vaccination, 23% neutral, and 12% concerned about adverse effects. This shift is attributed to positive vaccination experiences and emerging scientific data supporting vaccine safety [21].

VHL among university students is crucial for public health, as colleges are frequent sites of disease transmission [22]. University students are at an age where they make independent health decisions, influencing their peers and the wider community [23]. Given the prevalence of vaccine misinformation on social media, it is essential to equip students with accurate information and critical thinking skills [24]. Enhancing VHL can lead to better-informed decisions, promoting a culture of health and safety on campuses and beyond. The primary aim of our study is to investigate and analyze the attitudes and prejudices that students in Kazakhstan hold toward vaccines. By understanding these perspectives, we can better comprehend the underlying beliefs and misconceptions that may influence their views on vaccination.

## 2. Materials and Methods

This study utilized a cross-sectional survey design to explore the association between vaccine-related beliefs and attitudes and VHL scores among students throughout Kazakhstan. Data were collected between September and November of 2023 via a structured questionnaire distributed to a stratified random sample of students aged 18 years and older from diverse public higher education institutions across various regions. Out of the 3150 students surveyed, 3142 responded anonymously, resulting in a 99.75% response rate. Participants were enrolled in a wide range of educational programs, including Health Sciences, Humanities and Social Sciences, and Engineering. The research team strategically sent invitations to participate to universities in Kazakhstan’s major student hubs: Almaty, Astana, and Karaganda. These cities were selected due to their large and diverse student populations, making them ideal for capturing a representative sample. The invitations provided comprehensive information about the study, detailing its purpose, the health literacy (HL) assessment tool used, and the benefits of participation. These efforts ensured that potential participants were well-informed about the study’s scope and significance before deciding to participate. Administrative representatives from eight universities responded positively and agreed to collaborate. The universities’ administration organized the meetings between the research team and the students. During these meetings, researchers presented the objectives of the study in detail and explained the process for completing the questionnaire. This interactive approach helped to engage the students and clarify any questions they might have had. Students had the option to complete the questionnaire on paper or online via QR codes linked to Google Forms. Ensuring informed consent was a priority, and all participants provided their consent before participating in the study. The study received ethical approval from the Bioethics Committee of Karaganda Medical University (Approval No. 17, dated 20 September 2022), ensuring that all procedures were conducted in accordance with ethical standards.

### 2.1. VHL Assessment Tool

To assess students’ VHL, we used the HLS19-VAC instrument. This instrument is a 4-item questionnaire measuring VHL in the general adult population and is part of the HLS19 family of health literacy instruments. It was developed by the HLS19 (Health Literacy Survey 2019–2021) Consortium working group of 17 countries. HLS19 is the first project of the World Health Organization (WHO) Action Network on Measuring Public and Organizational Health Literacy (M-POHL; https://m-pohl.net (accessed on 7 August 2024)), coordinated by the HLS19 International Coordinating Centre (ICC) [25]. All items of the HLS19-VAC used a 4-point Likert scale with the following categories: “very easy”, “easy”, “difficult”, and “very difficult”. The VHL score was calculated as a range from 0 to 100 of questions answered “very easy” or “easy” among those with valid responses and was determined only for respondents who completed all four HLS19-VAC questions [26]. A higher score indicated a higher level of VHL.

### 2.2. VHL Correlates

In our study, we analyzed six vaccine-related belief and attitude measures (VHL correlates) as independent variables. This comprehensive package, referred to as HL-VAC, incorporated several additional components to provide a more nuanced understanding of VHL. The HL-VAC package included questions regarding personal vaccination behaviors over the past five years, assessing how frequently and consistently participants had received vaccinations. Furthermore, it included items measuring personal confidence in the efficacy and safety of vaccines, aiming to gauge the level of trust individuals have in the vaccination process. To address common misconceptions, the package also presented questions specifically designed to challenge widespread myths about potential vaccine risks [27]. Variables 1–5 utilized a 4-point scale ranging from Strongly Agree to Strongly Disagree. Due to limited variation within the positive (Strongly Agree and Agree) and negative (Strongly Disagree and Disagree) options, we combined these into binary variables: Agree and Disagree. Similarly, variable 6 (How high do you estimate the risk of developing a disease that you can be vaccinated against if you are not vaccinated?), which originally had a scale from Very High to Very Low, was converted to a binary variable (High vs. Low). The control variables included past behavior and demographic information. The past behavior question (Have you, your children, or has someone in your family had any vaccinations in the last five years?) required a simple Yes/No response.

### 2.3. Statistical Analysis

The dependent variable in our study was the vaccine health literacy (VHL) score, ranging from 0 (lowest) to 100 (highest). For the calculation of this score, please see The HLS19-VAC Instrument to measure Vaccination Literacy Factsheet [24]. The primary analytical method was Ordinary Least Squares (OLS) regression, which was used to assess the relationship between vaccine-related beliefs/attitudes and VHL, adjusting for confounding factors. Analysis was conducted using STATA SE 18, with a significance level set at *p* < 0.05.

## 3. Results

The descriptive statistics for the study variables are summarized in Table 1. The mean vaccine health literacy score among the students was 84.74 (SD = 27.99). Regarding vaccine-related beliefs and attitudes, a majority of students agreed with positive statements about vaccinations: 83.4% agreed that vaccinations were important to protect themselves and their children, 79.1% agreed that vaccinations were safe, 80.9% agreed that vaccinations were effective, 77.8% agreed that vaccination was compatible with their religious beliefs, and 87.8% agreed that vaccinations were important to prevent the spread of diseases. When assessing the perceived risk of developing a disease if not vaccinated, 68.3% of the students considered the risk to be high. For control variables, 89.0% reported having been vaccinated in the past five years. The average age of the participants was 20.16 years (SD = 4.84). In terms of gender, 56.1% were female, and 43.9% were male. Additionally, two-thirds of the students resided in urban areas, while one-third were from rural areas.

Figure 1 shows the VHL scores among students from different academic fields, as well as between those who had been vaccinated and those who had not. The syringe metaphor in the figure represents the VHL scores in various fields. The VHL scores were as follows: Engineering students scored 80.52, Humanities and Social Sciences students scored 85.16, and Health Sciences students had the highest score of 87.91. Overall, vaccinated students had a literacy score of 85.4, while those not vaccinated scored lower at 79.1.

Table 2 shows significant associations between VHL and all the vaccine-related beliefs and attitudes. Students who agreed that vaccinations were important to protect themselves and their children had a mean vaccine health literacy score of 88.2 (SD = 24.1), compared to 66.9 (SD = 38.1) for those who disagreed (*p* < 0.001). Those who believed vaccinations were safe had a mean score of 88.6 (SD = 23.9) versus 70.2 (SD = 36.2) for those who disagreed (*p* < 0.001). Similarly, students who agreed that vaccinations were effective had a higher mean score of 88.6 (SD = 24.0) compared to 68.6 (SD = 36.8) for those who disagreed (*p* < 0.001). Belief in the compatibility of vaccination with religious beliefs was also associated with higher literacy scores (87.9 [SD = 24.6] vs. 73.6 [SD = 35.3], *p* < 0.001). Students who agreed that vaccinations were important to prevent the spread of diseases had a mean score of 87.3 (SD = 25.2) versus 67.7 (SD = 38.2) for those who disagreed (*p* < 0.001). Additionally, those who perceived a high risk of developing a disease if not vaccinated had higher health literacy scores (87.2 [SD = 25.4] vs. 79.5 [SD = 32.3], *p* < 0.001).

Among control variables, students with past vaccination status had a higher mean vaccine health literacy score of 85.4 (SD = 27.2) compared to 79.1 (SD = 33.6) for those without past vaccinations (*p* < 0.001). Age showed a significant positive correlation with vaccine health literacy (r = 0.06, *p* < 0.001). Gender and location, however, were not significantly associated with vaccine health literacy, with mean scores for females (85.2 [SD = 27.2]) and males (84.2 [SD = 28.9]) showing no significant difference (*p* = 0.326) and rural (84.2 [SD = 27.8]) versus urban (85.0 [SD = 28.1]) also not showing a significant difference (*p* = 0.422).

The results of the regression analysis, examining the association between the independent variables and VHL while adjusting for potential confounders, are summarized in Table 3. The regression model indicated that positive beliefs and attitudes toward vaccinations were significantly associated with higher VHL scores. Specifically, students who agreed that vaccinations were important to protect themselves and their children had a significantly higher vaccine health literacy score (β = 8.774, *p* < 0.001) compared to those who disagreed. Similarly, those who believed vaccinations were safe (β = 5.863, *p* < 0.001) and effective (β = 7.250, *p* < 0.001) also had significantly higher scores.

The belief that vaccination was compatible with religious beliefs was also positively associated with vaccine health literacy (β = 3.333, *p* = 0.017). However, agreeing that vaccinations were important to prevent the spread of diseases did not show a significant association with vaccine health literacy (*p* = 0.438).

Regarding perceived risk, students who believed there was a high risk of developing a disease if not vaccinated had significantly higher vaccine health literacy scores compared to those who perceived a low risk (β = 2.403, *p* = 0.024).

Among control variables, past vaccination status (*p* = 0.151), age (*p* = 0.051), gender (*p* = 0.239), and location (*p* = 0.481) were not significantly associated with VHL.

## 4. Discussion

Our study found that most students had positive attitudes toward vaccinations, recognizing their importance for personal and child protection, safety, effectiveness, and compatibility with religious beliefs. Students who believed in these benefits had higher VHL. Health Sciences students had the highest VHL, followed by those in Humanities and Social Sciences, with Engineering students scoring the lowest. A majority of students had been vaccinated in the past five years, and those vaccinated generally had higher VHL. The belief in the high risk of disease without vaccination was linked to higher health literacy. Positive beliefs about vaccinations, such as their safety and effectiveness, were significantly correlated with higher VHL, while factors like gender and location did not show a significant association.

Although immunization “reaches more people than any other health or social service and is a vital component of primary health care” [28], attitudes toward vaccines and adherence to vaccination vary due to numerous influencing factors, with vaccination literacy being a particularly significant one. Research indicates that higher levels of vaccination literacy are associated with increased vaccine acceptance, which is consistent with our findings. Thus, Shon EJ et al. found that flu vaccine literacy has had a direct impact on flu vaccination, with individuals who possess a better understanding of vaccination being more likely to engage in immunization practices, such as getting vaccinated. Their research showed that flu vaccine literacy influenced health beliefs, including perceived benefits and the severity of the flu, which, in turn, affected flu vaccination rates. However, while beliefs in barriers acted as a mediator between vaccine literacy and flu vaccination, stronger beliefs in barriers significantly impeded vaccine uptake [29]. Moreover, it was demonstrated that vaccine literacy is a predictor of vaccination intention, while the association with vaccination status was marginally significant, indicating that other factors likely play a role in determining actual vaccination uptake [30]. Universities’ strategies for improving students’ adherence to vaccination should focus on increasing vaccination literacy and providing evidence-based information on vaccine safety and benefits. Research revealed that students who felt very well-informed about the COVID-19 vaccination were 8.47 times more likely to report being fully vaccinated compared to those who felt poorly informed. Additionally, students with a higher level of knowledge about COVID-19-related topics tended to have a deeper understanding of the pandemic’s complexity and were more likely to complete their vaccination regimen [31].

In the academic literature, there have been several studies conducted in Kazakhstan that explored attitudes and perceptions toward vaccination in adults recently [32,33,34]. However, it is important to note that these studies did not utilize a validated scale such as the HLS19-VAC to measure Vaccination HL. The HLS19-VAC is a validated tool designed to assess individuals’ knowledge, motivation, and skills in finding, understanding, evaluating, and applying immunization-related information [35]. University students, being future professionals, parents, and community leaders, are key to understanding vaccine hesitancy as their knowledge about vaccination significantly influences public health.

This study reveals significant associations between various vaccine-related beliefs and attitudes and VHL among students in Kazakhstan, underlining the impact of educational interventions on vaccination perceptions. Our findings demonstrated that students with positive attitudes toward vaccinations—deeming them safe, effective, and important for personal and public health—tended to have higher VHL scores. These results were consistent with previous research, indicating that enhanced health literacy can foster favorable vaccine-related beliefs and practices [36].

The positive correlation between VHL and the belief in the importance of vaccinations to protect oneself and one’s children suggests that personal relevance and perceived benefits of vaccinations are critical drivers of health literacy. This is aligned with the Health Belief Model, which posits that personal beliefs about health significantly influence health behavior [37]. Similarly, the associations between the safety and efficacy beliefs about vaccinations and higher VHL scores may reflect a well-informed student body, possibly due to effective health communication strategies within educational settings. However, while most beliefs were significantly associated with VHL, the belief that vaccinations prevent the spread of diseases did not show a statistically significant association. This could suggest that the concept of herd immunity may be less intuitive or less well-understood among students, indicating a potential area for educational enhancement.

The VHL scores among students in different fields of study and their vaccination status can provide insights into the understanding and attitudes toward vaccination. The scores in our study indicated that students in the Health Sciences field had the highest VHL at 87.91%, followed by students in the Humanities and Social Sciences at 85.16% and Engineering at 80.52%. This could be attributed to the nature of their studies, where Health Sciences students are more likely to be exposed to health-related information, including vaccination [38]. A significant part of the respondents (89%) was vaccinated, which is consistent with global efforts to increase vaccination coverage. The findings also revealed that individuals with a higher VHL score were vaccinated compared to those with lower HL levels. This suggests that individuals with a better understanding and processing of health information are more likely to engage in preventive health behaviors, aligning with previous research indicating that HL is a determinant of vaccination uptake [39]. Our study also revealed that individuals who hold positive attitudes toward vaccinations—agreeing that they are important, safe, and effective—demonstrated higher vaccination HL scores. Our findings align with existing research on the knowledge and attitudes of students regarding COVID-19 vaccinations.

Our analysis also highlighted the importance of perceived risks, with students who believed in the high risk of contracting diseases if unvaccinated showing higher VHL scores. This finding supports the notion that risk perception is a significant motivator for seeking health knowledge and adhering to preventive behaviors [40].

Control variables such as past vaccination status, age, gender, and location showed no significant associations with VHL. This might indicate that VHL is more strongly influenced by cognitive and attitudinal factors than demographic factors, suggesting that educational campaigns should focus more on addressing misconceptions and enhancing understanding rather than targeting specific demographic groups.

In Kazakhstan, as in many other countries, there are challenges related to vaccine hesitancy, which can be linked to the low VHL. One study identified that several factors, such as diminished trust in health authorities, lowered adherence to preventive measures and increased conspiracy beliefs, while a higher likelihood of vaccine acceptance was observed among adults with a greater trust in health information, adherence to preventive measures, and personal connections to COVID-19 impacts [41]. Another study focused on medical students in Kazakhstan found that having multiple vaccine choices could reduce vaccination hesitancy by about 30% [42]. This suggests that providing more options for vaccines could be a potential strategy to increase vaccine uptake. A 2020 study found that over a third of the 417 participants were hesitant about the COVID-19 vaccine, with hesitancy more common among women, those over 30, and parents. The vaccine’s country of origin significantly influenced this hesitancy [43]. According to UNICEF data, Kazakhstan has experienced a decline in the national vaccination rate for 15 routine vaccines, falling from 94.7% in 2018 to 89.1% in 2020, which could potentially affect the achievement of herd immunity necessary for preventing disease outbreaks [44]. A study by the Central Asia Regional Economic Cooperation (CAREC) found that while most respondents from the region expressed positive opinions about the effectiveness of vaccinations, the lowest proportion (29.2%) was found among respondents from Kazakhstan [20]. Moreover, in Kazakhstan’s transitional economy, a variety of risk factors affect childhood vaccine coverage, particularly high educational attainment among women and access to healthcare services [45].

Understanding the determinants of VHL is crucial for the successful implementation of intervention strategies. Our study’s regression model reveals that several key factors significantly influence higher VHL rates.

First, the belief in the importance of vaccination for protection against disease emerged as a critical determinant. Individuals who perceive vaccines as essential for preventing illness are more likely to be knowledgeable about vaccination.

Second, agreement on the safety and effectiveness of vaccines was another significant factor. Trust in the safety and efficacy of vaccines directly correlated with higher literacy levels, suggesting that educational campaigns should focus on disseminating evidence-based information about vaccine safety to build public confidence.

The compatibility of vaccination with religious beliefs also played a notable role. Our findings indicated that when vaccination practices do not conflict with religious values, individuals tend to exhibit higher VHL. This suggests that engaging religious leaders and framing vaccination in a context that respects religious beliefs could enhance literacy and acceptance.

Lastly, the belief that not vaccinating increases the risk of disease significantly contributed to VHL. This highlighted the importance of conveying the risks associated with non-vaccination to emphasize the protective benefits of vaccines. Educational interventions should thus include information on the potential consequences of avoiding vaccination to reinforce the perceived necessity of vaccines.

Age was also a determinant of VHL, as in the study by Cadeddu et al. The findings from Cadeddu et al.‘s study highlighted the importance of considering age-related factors when designing educational interventions aimed at improving VHL across different age groups. [46]

Collectively, these determinants underscored the importance of addressing both personal and contextual beliefs to enhance VHL. Tailored communication strategies that consider individual beliefs and cultural contexts are essential for improving VHL and, consequently, vaccination rates. Future interventions should prioritize these factors to effectively increase public understanding and acceptance of vaccines.

Individuals with higher health literacy are more likely to understand vaccine information, increasing their likelihood of getting vaccinated. Educational interventions that address misconceptions can improve students’ attitudes toward vaccination [47]. Hesitancy often arises from negative media reports and distrust in vaccine companies [48,49]. Anti-vaccine content on social media, which generates high engagement, underscores the need for targeted public health campaigns and accurate information from trusted sources to counteract misinformation [50]. Peer-based education and open dialogues with healthcare professionals can help build trust in vaccines [51,52]. These educational efforts should adhere to human rights principles and be tailored to Kazakhstan’s specific context, sometimes necessitating restrictions on individual freedoms to protect public health [53,54,55].

This study highlighted the importance of conveying the risks associated with non-vaccination to enhance VHL among students. The finding that understanding these risks significantly contributes to VHL aligns with Kazakhstan’s recent initiatives in public health. For instance, the government’s decision to implement HPV vaccination for girls in September 2024 reflects an awareness of both scientific evidence and public perception [56,57,58]. To ensure the success of this program, educational interventions should emphasize the potential consequences of avoiding vaccination, thereby reinforcing the perceived necessity of vaccines. This approach complements Kazakhstan’s comprehensive preparation efforts, including the incorporation of the HPV vaccine into the national immunization schedule and the engagement of young immunization activists [59]. These initiatives, particularly the ‘Youth in Action’ campaign, provide an excellent platform to address vaccination literacy gaps identified in this study, potentially increasing vaccine acceptance and uptake among the student population [60].

### 4.1. Strengths

One of the key strengths of this study is the large and diverse sample size, which included students from various regions and types of higher education institutions across Kazakhstan. This enhances the generalizability of the findings to the broader student population in the country. Additionally, the use of a validated scale to measure vaccine health literacy and the comprehensive approach to data collection, which included both demographic and contextual information, allowed for a thorough analysis of the factors associated with vaccine health literacy.

### 4.2. Limitations

Despite its strengths, this study has several limitations. The cross-sectional design limited the ability to infer causality between vaccine-related beliefs and attitudes and VHL. Longitudinal studies are needed to establish temporal relationships and causality. Another limitation is the potential for self-reporting bias, as the data were collected through self-administered questionnaires. Students may have provided socially desirable responses rather than their true beliefs and behaviors.

The study also did not account for all possible confounding variables. Factors such as socioeconomic status, access to healthcare, and previous experiences with the healthcare system, which can significantly influence health literacy and attitudes toward vaccination, were not included. Finally, the binary categorization of several independent variables might have oversimplified complex attitudes and beliefs, potentially obscuring more nuanced associations.

## 5. Conclusions

Vaccination is currently recognized as one of the most effective and economical public health measures globally. The results of this study underscore the significant association between positive vaccine-related beliefs and higher vaccine health literacy among students. A large majority of students expressed agreement with positive statements about vaccinations, including their importance, safety, effectiveness, and compatibility with religious beliefs. These beliefs, along with the perceived high risk of disease if not vaccinated, were strongly correlated with higher literacy scores. While past vaccination status and age showed some correlation with VHL, gender, and location did not significantly impact literacy scores. Our findings are relevant to ongoing public health initiatives in Kazakhstan, such as the government’s decision to implement HPV vaccination for girls starting September 2024. Understanding students’ knowledge and attitudes toward vaccines can contribute to the success of such programs. Our findings also highlighted the need for targeted educational interventions that reinforce positive vaccine beliefs to improve VHL and thereby potentially increase vaccination rates among students.

## Figures and Tables

**Figure 1 vaccines-12-00917-f001:**
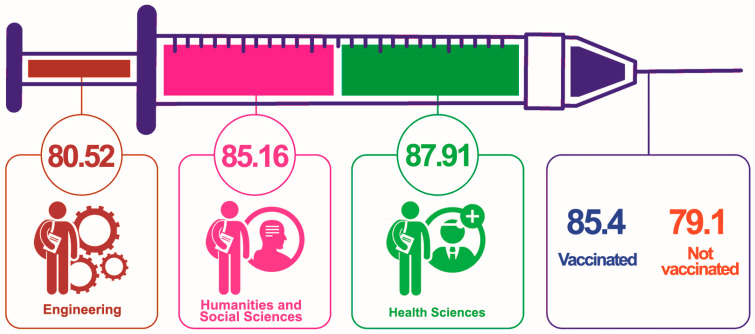
Mean VHL score among students of different educational programs and vaccination status.

**Table 1 vaccines-12-00917-t001:** Descriptive statistics of the sample (N = 3142).

Variables	Mean (SD)/Frequency (%)
**Dependent variable**	
Vaccine health literacy [mean (SD)]	84.74 (27.99)
**Independent variables**	
**Vaccine-related beliefs and attitude**	
Vaccinations are important to protect myself and my children	
Disagree	514 (16.4%)
Agree	2628 (83.4%)
Vaccinations are safe	
Disagree	658 (20.9%)
Agree	2484 (79.1%)
Vaccinations are effective	
Disagree	600 (19.1%)
Agree	2542 (80.9%)
Vaccination is compatible with my religious beliefs	
Disagree	698 (22.2%)
Agree	2444 (77.8%)
Vaccinations are important to prevent the spread of diseases	
Disagree	406 (12.9%)
Agree	2736 (87.8%)
If not vaccinated, the risk of developing a disease is	
Low	998 (31.7%)
High	2147 (68.3%)
**Control variables**	
**Behavioral information**	
Past vaccination status	
No	361 (11.0%)
Yes	2905 (89.0%)
**Demographic information**	
Age [mean (SD)]	20.16 (4.84)
Gender	
Female	1762 (56.1%)
Male	1380 (43.9%)
Location	
Rural	1018 (32.4%)
Urban	2124 (67.6%)

**Table 2 vaccines-12-00917-t002:** Bivariate analysis between health literacy and other variables (N = 3142).

Variables	Vaccine Health Literacy	*p*-Value
**Independent variables**		
**Vaccine-related attitude information**		
Vaccinations are important to protect myself and my children		
Disagree	66.9 (38.1)	<0.001
Agree	88.2 (24.1)	
Vaccinations are safe		
Disagree	70.2 (36.2)	<0.001
Agree	88.6 (23.9)	
Vaccinations are effective		
Disagree	68.6 (36.8)	<0.001
Agree	88.6 (24.0)	
Vaccination is compatible with my religious beliefs		
Disagree	73.6 (35.3)	<0.001
Agree	87.9 (24.6)	
Vaccinations are important to prevent the spread of diseases		
Disagree	67.7 (38.2)	<0.001
Agree	87.3 (25.2)	
If not vaccinated, the risk of developing a disease is		
Low	79.5 (32.3)	<0.001
High	87.2 (25.4)	
**Control variables**		
**Behavioral information**		
Past vaccination status		
No	79.1 (33.6)	<0.001
Yes	85.4 (27.2)	
**Demographic information**		
Age	0.06	<0.001
Gender		
Female	85.2 (27.2)	0.326
Male	84.2 (28.9)	
Location		
Rural	84.2 (27.8)	0.422
Urban	85.0 (28.1)	

**Table 3 vaccines-12-00917-t003:** Regression analysis (N = 3142).

	Beta Coefficient	95% Confidence Intervals		*p*-Value
Vaccinations are important to protect myself and my children				
Disagree	ref			
Agree	8.774	5.115	12.434	<0.001
Vaccinations are safe				
Disagree	ref			
Agree	5.863	2.569	9.157	<0.001
Vaccinations are effective				
Disagree	ref			
Agree	7.250	3.620	10.880	<0.001
Vaccination is compatible with my religious beliefs				
Disagree	ref			
Agree	3.333	0.606	6.061	0.017
Vaccinations are important to prevent the spread of diseases				
Disagree	ref			
Agree	1.523	−2.329	5.374	0.438
If not vaccinated, the risk of developing a disease is				
Low	ref			
High	2.403	0.317	4.490	0.024
**Control variables**				
**Behavioral information**				
Past vaccination status				
No				
Yes	2.211	−0.811	5.231	0.151
**Demographic information**				
Age	0.192	−0.001	0.385	0.051
Gender				
Male	ref			
Female	1.144	−0.759	3.048	0.239
Location				
Urban	ref			
Rural	−0.719	−2.716	1.279	0.481
Constant	55.085	49.876	60.294	<0.001

## Data Availability

The data presented in this study are available on request from the corresponding author.

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
