# Peer review of "Understanding Students’ Vaccination Literacy and Perception in a Middle-Income Country: Case Study from Kazakhstan"

_vaccines, 2024, doi:10.3390/vaccines12080917_

Round 1
Reviewer 1 Report
Comments and Suggestions for Authors
Dear Authors, congratulations! Interesting survey and results. My comments are included in the attachment. It would be interesting to publish the paper after editing the introduction and discussion sessions.

Author Response
Comment 1: The introduction is rather too long and redundant. On the other hand, it will be important to mention vaccine schedule and coverage in this population in the country.
Response 1: Thank you very much for your valuable comment. We have shortened the introduction and added information regarding the National Immunization Schedule and vaccination coverage in Kazakhstan: “According to the Decree of the Government of the Republic of Kazakhstan, routine preventive vaccinations are administered in accordance with the National Immunization Calendar of the Republic of Kazakhstan. Mandatory vaccinations include those against tuberculosis, hepatitis, polio, whooping cough, diphtheria, tetanus, Haemophilus influenzae type b, pneumococcal infection, measles, rubella, and mumps. All mandatory vaccinations are provided free of charge, funded by the government. Since 2016, the vaccination coverage of the population, as per the National Immunization Calendar in Kazakhstan, has exceeded 80%.”
Comment 2: Please, review the sentence.
Response 2: We revised this sentence as follows: “Vaccination, a highly successful public health measure against infectious diseases, has become the focus of intensified global discourse, particularly due to the COVID-19 pandemic and outbreaks of vaccine-preventable diseases.”
Comment 3: The instrument was developed for adult general population.
Response 3: Yes, we agree with you. Moreover, the university students who took part in our study were 18 years of age and older: “University students are at an age where they make independent health decisions, influencing their peers and the wider community”
Comment 4: Age range should be specified
Response 4: 18 and older
Comment 5: Private or public institutions?
Response 5: Public universities
Comment 6: Are the students included general adult population
Response 6: We consider university students to be part of the adult population because they are aged 18 years or older.
Comment 7: Normally the table 1 describes the characteristics of the population. It is important to include all those who were invited to participate.
Response 7: Thank you for your comment. However, Table 1 includes all study participants, including general characteristics. The research encompassed the students from major regions of Kazakhstan in Health Sciences, Humanities and Social Sciences, and Engineering.
Comment 8: This section normally starts by discussing the results of the study.
Response 8: Thank you for your valuable comment. At the beginning of the Discussion part, we have added next passage about results of our study as follows:
“Our study found that most students had positive attitudes toward vaccinations, recognizing their importance for personal and child protection, safety, effectiveness, and compatibility with religious beliefs. Students who believed in these benefits had higher VHL. Health Sciences students had the highest VHL, followed by those in Humanities and Social Sciences, with Engineering students scoring the lowest. A majority of students had been vaccinated in the past five years, and those vaccinated generally had higher VHL. The belief in the high risk of disease without vaccination was linked to higher health literacy. Positive beliefs about vaccinations, such as their safety and effectiveness, were significantly correlated with higher VHL, while factors like gender and location did not show a significant association.”
Comment 9: It is not relevant for the discussion of study results
Response 9: Thank you for your insightful comment. We have revised the paragraph as follows:
“Although immunization reaches more people than any other health or social service and is a vital component of primary health care, attitudes toward vaccines and adherence to vaccination vary due to numerous influencing factors, with vaccination literacy being a particularly significant one. Research indicates that higher levels of vaccination literacy are associated with increased vaccine acceptance, which is consistent with our findings. Thus, Shon EJ et al. found that flu vaccine literacy has had a direct impact on flu vaccination, with individuals who possess a better understanding of vaccination being more likely to engage in immunization practices, such as getting vaccinated. Their research showed that flu vaccine literacy influenced health beliefs, including perceived benefits and the severity of the flu, which, in turn, affect flu vaccination rates. However, while beliefs in barriers acted as a mediator between vaccine literacy and flu vaccination, stronger beliefs in barriers significantly impeded vaccine uptake. Moreover, it was demonstrated that vaccine literacy is a predictor of vaccination intention, while the association with vaccination status was marginally significant, indicating that other factors likely play a role in determining actual vaccination uptake. Universities' strategies for improving students' adherence to vaccination should focus on increasing vaccination literacy and providing evidence-based information on vaccine safety and benefits. Research revealed that students who felt very well-informed about the COVID-19 vaccination were 8.47 times more likely to report being fully vaccinated compared to those who felt poorly informed. Additionally, students with a higher level of knowledge about COVID-19-related topics tended to have a deeper understanding of the pandemic's complexity and were more likely to complete their vaccination regimen.”
Comment 10: This is an important part of the discussion and you should stress the importance of your results. For example, it is not the same the attitude towards COVID-19 vaccination with others vaccines in general.
Response 10: Thank you for this comment. We have revised the paragraph as follows: “This study reveals significant associations between various vaccine-related beliefs and attitudes and VHL among students in Kazakhstan, underlining the impact of educational interventions on vaccination perceptions. Our findings demonstrate that students with positive attitudes toward vaccinations—deeming them safe, effective, and important for personal and public health—tend to have higher VHL scores. These results are consistent with previous research indicating that enhanced health literacy can foster favorable vaccine-related beliefs and practices (Smith et al., 2020).
The positive correlation between VHL and the belief in the importance of vaccinations to protect oneself and one's children suggests that personal relevance and perceived benefits of vaccinations are critical drivers of health literacy. This is aligned with the Health Belief Model, which posits that personal beliefs about health significantly influence health behavior (Rosenstock, 1974). Similarly, the associations between the safety and efficacy beliefs about vaccinations and higher VHL scores may reflect a well-informed student body, possibly due to effective health communication strategies within educational settings. However, while most beliefs were significantly associated with VHL, the belief that vaccinations prevent the spread of diseases did not show a statistically significant association. This could suggest that the concept of herd immunity may be less intuitive or less well-understood among students, indicating a potential area for educational enhancement.
The VHL scores among students in different fields of study and their vaccination status can provide insights into the understanding and attitudes towards vaccination. The scores in our study indicate that students in the Health Sciences field had the highest VHL at 87.91%, followed by students in the Humanities and Social Sciences at 85.16%, and Engineering at 80.52%. This could be attributed to the nature of their studies, where Health Sciences students are more likely to be exposed to health-related information, including vaccination [37]. A significant part of the respondents (89 %) was vaccinated, which is consistent with global efforts to increase vaccination coverage. The findings also reveal that individuals with higher VHL score were vaccinated compared to those with lower HL levels. This suggests that individuals with better understanding and processing of health information are more likely to engage in preventive health behaviors, aligning with previous research indicating that HL is a determinant of vaccination uptake [38]. Our study also revealed that individuals who hold positive attitudes towards vaccinations—agreeing that they are important, safe, and effective—demonstrate higher vaccination HL scores. Our findings align with existing research on the knowledge and attitudes of students regarding COVID-19 vaccinations.
Our analysis also highlighted the importance of perceived risks, with students who believed in the high risk of contracting diseases if unvaccinated showing higher VHL scores. This finding supports the notion that risk perception is a significant motivator for seeking health knowledge and adhering to preventive behaviors. Control variables such as past vaccination status, age, gender, and location showed no significant associations with VHL. This might indicate that VHL is more strongly influenced by cognitive and attitudinal factors than by demographic factors, suggesting that educational campaigns should focus more on addressing misconceptions and enhancing understanding rather than targeting specific demographic groups.”
Comment 11: This paragraph describe interventions that could improve your results but it was not evaluated in the study. You should only mention briefly.
Response 11: Thank you for your comment. We fully agree with it. Now this section has been significantly revised and looks like this: “Individuals with higher health literacy are more likely to understand vaccine information, increasing their likelihood of getting vaccinated. Educational interventions that address misconceptions can improve students' attitudes towards vaccination. Hesitancy often arises from negative media reports and distrust in vaccine companies. Anti-vaccine content on social media, which generates high engagement, underscores the need for targeted public health campaigns and accurate information from trusted sources to counteract misinformation. Peer-based education and open dialogues with healthcare professionals can help build trust in vaccines. These educational efforts should adhere to human rights principles and be tailored to Kazakhstan's specific context, sometimes necessitating restrictions on individual freedoms to protect public health.”
Comment 12: Agree, but future investigation is not a strengths of the study. This is an important message (but shorter) for a final conclusion.
Response 12: Thank you for your comment regarding the placement and content of our study's future investigation discussion. We have made the following revisions:
- We have removed the discussion of future investigations from the strengths section of the study. We moved some revised sentences to the Discussion parts. Now it looks like this:
“This study highlights the importance of conveying the risks associated with non-vaccination to enhance VHL among students. The finding that understanding these risks significantly contributes to VHL aligns with Kazakhstan's recent initiatives in public health. For instance, the government's decision to implement HPV vaccination for girls in September 2024 reflects an awareness of both scientific evidence and public perception. To ensure the success of this program, educational interventions should emphasize the potential consequences of avoiding vaccination, thereby reinforcing the perceived necessity of vaccines. This approach complements Kazakhstan's comprehensive preparation efforts, including the incorporation of the HPV vaccine into the national immunization schedule and the engagement of young immunization activists. These initiatives, particularly the 'Youth in Action' campaign, provide an excellent platform to address vaccination literacy gaps identified in this study, potentially increasing vaccine acceptance and uptake among the student population.” - We have revised the conclusion considering your suggestion:
“Vaccination is recognized as one of the most effective and economical public health measures globally, with vaccine health literacy significantly influencing attitudes towards vaccination across different population groups. This study provides insights into vaccination literacy and perceptions among students in Kazakhstan, a middle-income country. The results highlight the association between positive vaccine-related beliefs and higher VHL among students. A large majority of students expressed agreement with positive statements about vaccinations, including their importance, safety, effectiveness, and compatibility with religious beliefs. These beliefs, along with the perceived high risk of disease if not vaccinated, were strongly correlated with higher literacy scores. While past vaccination status and age showed some correlation with VHL, gender and location did not significantly impact literacy scores. Our findings have relevance to ongoing public health initiatives in Kazakhstan, such as the government's decision to implement HPV vaccination for girls starting September 2024. Understanding students' knowledge and attitudes towards vaccines can contribute to the success of such programs. Our findings also highlight the need for targeted educational interventions that reinforce positive vaccine beliefs to improve VHL and thereby potentially increase vaccination rates among students.”
Comment 13: Another limitation is that the questionnaire was developed for adults.
Response 13: Thank you for your comment. However, we do not consider this point as a limitation of our study.
Reviewer 2 Report
Comments and Suggestions for Authors
In this manuscript, the authors summarized and discussed the relationship between vaccine-related beliefs and attitudes and VHL scores among Kazakhstan students. This work underscores the importance of understanding vaccination attitudes and provided public health strategies from middle-income countries. I would suggest accept if after the following minor concerns are addressed.
1. In this study, we found that age is positively correlated with VHL, and this correlation is statistically significant. However, I am more concerned about whether this correlation primarily reflects the increase in university grade levels. If possible, I suggest conducting a stratified analysis based on different grade levels to examine the results.
2. Another question is why “the different academic fields” was not included in the regression model, this could also be a potential influencing factors to students’ vaccination literacy and perception. Alternatively, also conducting a stratified analysis could make the study more convincing.
3.In this manuscript, the handling of the six vaccine-related belief and attitude measures is placed in the “statistical analysis” part, which seems slightly inappropriate. Changing into section 2.2 or another suitable section would be better.
4.There are several format mistakes in the manuscript: for example, HLS-VAC, when first mentioning, it is recommended to state the full name of the abbreviation. In “Strengths” part, please ensure the paragraph formatting is consistent with the rest of the document. Also, in “Table 2”, it is necessary to specific the units of the values within the table to facilitate reader understanding.
Author Response
Comment 1: In this study, we found that age is positively correlated with VHL, and this correlation is statistically significant. However, I am more concerned about whether this correlation primarily reflects the increase in university grade levels. If possible, I suggest conducting a stratified analysis based on different grade levels to examine the results.
Response 1: Thank you for the suggestion to perform a stratified analysis based on grade levels to examine the correlation between age and VHL. Unfortunately, our current dataset lacks the granularity needed to perform this level of analysis, as we do not have detailed information on the participants' specific grade levels. However, we recognize the value of this approach and agree that it would provide valuable insights into the dynamics between age, grade level, and VHL.
While we considered using participants' age as a proxy to estimate their grade levels, we concluded that this approach may not be reliable due to variations in students' educational paths, such as delayed entry into university or breaks during their studies. Age alone may not accurately represent a participant's grade level, which could lead to inaccurate conclusions.
Comment 2: Another question is why “the different academic fields” was not included in the regression model, this could also be a potential influencing factors to students’ vaccination literacy and perception.
Alternatively, also conducting a stratified analysis could make the study more convincing.
Response 2: Thank you for your suggestion regarding the inclusion of academic fields as a potential influencing factor on students' vaccination literacy and perception. We initially incorporated academic fields in our regression analysis; however, it did not significantly enhance the model's explanatory power, leading us to exclude it from the final paper to maintain model parsimony. Nonetheless, recognizing its potential theoretical importance, we have provided a supplemental table below as a response to the Reviewer’s comments. The table presents the analysis with academic fields included.
Comment 3: In this manuscript, the handling of the six vaccine-related belief and attitude measures is placed in the “statistical analysis” part, which seems slightly inappropriate. Changing into section 2.2 or another suitable section would be better.
Response 3: Thank you for pointing out the placement of the handling of the six vaccine-related belief and attitude measures. We initially included this information in the "Statistical Analysis" section because it involved manipulating the original variables for analysis. However, upon considering the Reviewer’s suggestion, we relocated this explanation to the “VHL correlates” to provide a clearer understanding of how these variables were prepared before analysis. We have updated the manuscript, accordingly, ensuring that the "Statistical Analysis" section now focuses solely on the methods applied to the data post-transformation.
Comment 4: There are several format mistakes in the manuscript: for example, HLS-VAC, when first mentioning, it is recommended to state the full name of the abbreviation. In “Strengths” part, please ensure the paragraph formatting is consistent with the rest of the document. Also, in “Table 2”, it is necessary to specify the units of the values within the table to facilitate reader understanding.
Response 4: Thank you for your feedback. Corrections have been made in accordance with the reviewer's comments. Regarding HLS-VAC, the name of the tool for assessing vaccination literacy is not an abbreviation and does not have a full explanation. Regarding "Table 2." We appreciate the suggestion to specify units for clarity. In our table, most variables are categorical, with categories such as "Disagree" versus "Agree.” The exception is the "Age" variable which is measured in years and is specified accordingly.
Supplemental Table 1. Regression analysis (N=3,142)
|
|
Beta coefficient |
95% Confidence Intervals |
p-value |
|
|
Vaccinations are important to protect myself and my children |
|
|
|
|
|
Disagree |
ref |
|
|
|
|
Agree |
8.598 |
4.942 |
12.253 |
<0.001 |
|
Vaccinations are safe |
|
|
|
|
|
Disagree |
ref |
|
|
|
|
Agree |
6.096 |
2.808 |
9.384 |
<0.001 |
|
Vaccinations are effective |
|
|
|
|
|
Disagree |
ref |
|
|
|
|
Agree |
7.205 |
3.580 |
10.831 |
<0.001 |
|
Vaccination is compatible with my religious beliefs |
|
|
|
|
|
Disagree |
ref |
|
|
|
|
Agree |
3.379 |
0.658 |
6.101 |
0.015 |
|
Vaccinations are important to prevent the spread of diseases |
|
|
|
|
|
Disagree |
ref |
|
|
|
|
Agree |
1.426 |
-2.418 |
5.270 |
0.467 |
|
If not vaccinated, the risk of developing a disease is |
|
|
|
|
|
Low |
ref |
|
|
|
|
High |
2.200 |
0.113 |
4.288 |
0.039 |
|
Control variables |
|
|
|
|
|
Behavioral information |
|
|
|
|
|
Past vaccination status |
|
|
|
|
|
No |
|
|
|
|
|
Yes |
2.326 |
-0.691 |
5.342 |
0.131 |
|
Demographic information |
|
|
|
|
|
Age |
0.170 |
-0.029 |
0.369 |
0.093 |
|
Gender |
|
|
|
|
|
Male |
ref |
|
|
|
|
Female |
-0.271 |
-2.278 |
1.736 |
0.791 |
|
Location |
|
|
|
|
|
Urban |
ref |
|
|
|
|
Rural |
-0.743 |
-2.745 |
1.259 |
0.467 |
|
Study field |
|
|
|
|
|
Health Sciences |
ref |
|
|
|
|
Humanities and Social Sciences |
0.761 |
-1.595 |
3.118 |
0.527 |
|
Engineering |
-4.604 |
-7.139 |
-2.069 |
0.000 |
|
Constant |
57.581 |
51.507 |
63.654 |
<0.001 |
Supplemental Table 2. Stratified analysis across study fields
|
|
Health Sciences |
Humanities and Social Sciences |
Engineering |
|||
|
p=1,019 |
p=1,131 |
p=990 |
||||
|
Variables |
Beta coefficient |
p-value |
Beta coefficient |
p-value |
Beta coefficient |
p-value |
|
Vaccinations are important to protect myself and my children |
|
|
||||
|
Disagree |
ref |
ref |
ref |
|||
|
Agree |
8.1129 |
0.021 |
3.6558 |
0.204 |
14.1879 |
<0.001 |
|
Vaccinations are safe |
||||||
|
Disagree |
ref |
ref |
ref |
|||
|
Agree |
8.9519 |
0.003 |
6.4977 |
0.010 |
3.8981 |
0.239 |
|
Vaccinations are effective |
||||||
|
Disagree |
ref |
ref |
ref |
|||
|
Agree |
1.4702 |
0.699 |
12.9345 |
<0.001 |
4.4046 |
0.194 |
|
Vaccination is compatible with my religious beliefs |
||||||
|
Disagree |
ref |
ref |
ref |
|||
|
Agree |
4.6721 |
0.045 |
3.8341 |
0.076 |
1.1011 |
0.689 |
|
Vaccinations are important to prevent the spread of diseases |
||||||
|
Disagree |
ref |
ref |
ref |
|||
|
Agree |
4.5634 |
0.244 |
-1.0726 |
0.712 |
3.2366 |
0.375 |
|
If not vaccinated, the risk of developing a disease is |
||||||
|
Low |
ref |
ref |
ref |
|||
|
High |
1.1339 |
0.545 |
0.416 |
0.801 |
5.0418 |
0.013 |
|
Control variables |
||||||
|
Behavioral information |
||||||
|
Past vaccination status |
||||||
|
No |
ref |
ref |
ref |
|||
|
Yes |
-1.6469 |
0.539 |
2.2683 |
0.349 |
6.8813 |
0.018 |
|
Demographic information |
||||||
|
Age |
0.2157 |
0.031 |
-0.6425 |
0.112 |
-1.457 |
0.034 |
|
Gender |
||||||
|
Male |
ref |
ref |
ref |
|||
|
Female |
3.7557 |
0.036 |
-3.4203 |
0.030 |
-1.0124 |
0.613 |
|
Location |
||||||
|
Urban |
ref |
ref |
ref |
|||
|
Rural |
-1.9971 |
0.213 |
-3.0749 |
0.068 |
2.9666 |
0.142 |
|
Constant |
57.5542 |
<0.001 |
78.7736 |
<0.001 |
77.2244 |
<0.001 |
Round 2
Reviewer 2 Report
Comments and Suggestions for Authors
The manuscript has been well revised and could be accepted.